# Improving Batch Normalization in Federated Learning with Non-IID Features

## Abstract

Batch normalization (BN) has become a standard practice in deep neural networks, offering significant advantages in convergence speed and training stability. Recent studies indicate that applying BN in non-IID federated learning (FL) scenarios may result in performance degradation. However, these studies either replace it with alternative normalization methods or incur significant communication overhead. Moreover, they focus primarily on the non-IID labels scenario, neglecting the impact of non-IID features on BN in FL. In this paper, we carefully examine the challenges of BN under feature-shift FL and aim to retain BN in the model, which is crucial for leveraging abundant pretrained backbones. We revisit the FL procedure and show that BN statistics can vary significantly across clients in the presence of non-IID features, resulting in a notable train–test inconsistency. To address this issue, we propose Local–Global Consistency Regularization (GReg), a simple method that achieves alignment with global statistics during local training through an additional KL-based regularization term. Extensive experiments on three natural image benchmarks and a medical image benchmark demonstrate that GReg consistently improves FedAvg and several state-of-the-art FL methods. In addition, GReg exhibits strong generalization to unseen clients, works across diverse CNN and ViT-style architectures, and is suitable for both cross-silo and cross-device FL settings.

## 1 Introduction

Federated learning (FL) (McMahan et al., 2017) has garnered significant attention in recent years as a decentralized approach to machine learning, allowing the model to be trained across multiple clients while keeping data private. Despite its advantages in preserving data privacy and security, FL faces the challenge of heterogeneous data (Zhao et al., 2018). In real-world FL, different clients collect data in diverse environments and contexts, leading to variations in data distribution. The non-independently and identically distributed (non-IID) client data causes complications in the model training, leading to issues such as biased local model updates and slower global model convergence (Li et al., 2020; Karimireddy et al., 2020).

Batch normalization (BN) (Ioffe & Szegedy, 2015; Luo et al., 2018) has been a fundamental component of deep neural networks since its introduction, offering benefits such as mitigating internal covariate shift, accelerating convergence, and improving generalization. Although alternatives like layer normalization (LN) have gained prominence in Transformer architectures (Vaswani et al., 2017), BN remains the dominant choice in convolutional neural networks (CNNs), which are more lightweight and thus better suited for FL than Transformer models. Furthermore, many widely-used pretrained vision models are built on CNN backbones with BN (He et al., 2016; Huang et al., 2017), offering natural initialization for FL and enabling faster convergence. However, recent studies (Hsieh et al., 2020; Wang et al., 2023) reveal that BN can suffer in FL under non-IID client data, which is attributed to external covariate shifts between clients (Du et al., 2022).

To address the performance degradation caused by BN, existing FL works (Wang et al., 2023; Hsieh et al., 2020; Du et al., 2022; Zhong et al., 2023; Guerraoui et al., 2024; Chen et al., 2025; Wang et al., 2025) mainly focus on the label-shift FL, where the label distributions vary across clients. An equally important but less explored setting is **feature-shift FL** (Li et al., 2021b), where clients' data come from different domains (Wang et al., 2024), resulting in non-IID features. For example,

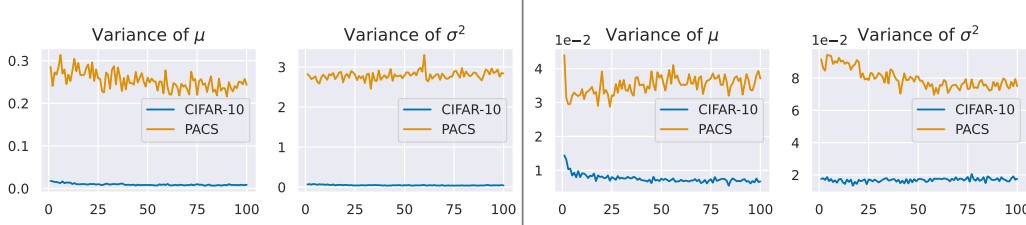

Figure 1: Variance of the BN statistics across clients versus communication rounds on label non-IID *CIFAR-10* and *PACS*. The two plots on the left correspond to a shallow layer of ResNet-18, while the two on the right correspond to a deep layer.

for image data, variations in lighting, style, and background can cause noticeable shifts in feature distributions. A representative dataset is the *PACS* (Li et al., 2017), where the same object categories appear as photos, artistic paintings, cartoons, and sketches, exhibiting strong style shifts. As BN operates by normalizing the intermediate features and accumulating their statistics, shifts in the feature distribution across clients directly affect the collected statistics, as also observed in prior work on domain adaptation (Li et al., 2016). In contrast, non-IID labels only impact BN indirectly through changes in the learned representations. Thus, feature-shift FL is more challenging than label-shift FL.

To further illustrate this problem, we perform the following experiment[1] on datasets *PACS* (with non-IID features) and *CIFAR-10* (with non-IID labels).[2] When evaluating the global model trained by FedAvg (McMahan et al., 2017), we use the test-time batch statistics instead of the global statistics aggregated across clients throughout federated training. One

Table 1: Test accuracies (%) of a FedAvg-trained global model on *PACS* and *CIFAR-10*, evaluated using either global statistics or batch statistics.

| Method | PACS | CIFAR-10 |
|---|---|---|
| Global Stats | 89.8 | 91.4 |
| Batch Stats | 90.9 (+1.1%) | 86.2 (-5.2%) |

would expect global statistics to perform better, as this aligns with the standard practice in centralized training. However, as can be seen from Table 1, while the use of batch statistics causes an accuracy drop of 5.2% on *CIFAR-10*, its accuracy on *PACS* is improved by 1.1%. This difference is further illustrated in Figure 1, which shows that the BN statistics vary much more widely among clients for *PACS* than for *CIFAR-10*. The significantly larger variance implies a greater mismatch between the local and global statistics, making feature-shift FL more difficult than label-shift FL.

Motivated by the above intriguing observations, in this paper, we investigate why BN fails in feature-shift FL. As will be discussed in Section 3, the main issue of BN in feature-shift FL is that activations normalized by the global statistics correspond to an unseen distribution for the global model, creating a train–test inconsistency. To alleviate this problem, we propose a simple yet effective solution, **Local–Global Consistency Regularization (GReg)**, which explicitly aligns the predictions produced under batch statistics and global statistics during local training. By introducing a lightweight regularization term based on KL divergence, GReg reduces the mismatch without changing the network architecture or incurring any extra communication. We evaluate GReg by integrating it with existing FL methods. Extensive experiments on datasets with non-IID features show that GReg consistently improves feature-shift FL methods, highlighting its effectiveness and applicability.

## 2 RELATED WORK

### 2.1 BATCH NORMALIZATION

Given a batch $b$ of activations $\{x\}$, Batch Normalization (BN) (Ioffe & Szegedy, 2015) standardizes them using the batch mean $\mu_b$ and variance $\sigma_b^2$, followed by an affine transformation with learnable

---

[1]Details are in Appendix A.

[2]Following (Gao et al., 2022; Acar et al., 2021), we construct heterogeneous label distributions for *CIFAR-10* by sampling from a Dirichlet distribution $\text{Dir}(\alpha)$ with $\alpha = 0.3$.

channel-wise parameters $\gamma$ and $\beta$:

$$BN(x) = \frac{x - \mu_b}{\sqrt{\sigma_b^2 + \epsilon}}\gamma + \beta, \tag{1}$$

where $\epsilon$ is a small value for numerical stability. Each BN layer also maintains running statistics $\bar{\mu}$ and $\bar{\sigma}^2$, which are updated in every batch via an exponential moving average:

$$\bar{\mu} \quad \leftarrow \quad (1 - \rho)\bar{\mu} + \rho\mu_b, \tag{2}$$
$$\bar{\sigma}^2 \quad \leftarrow \quad (1 - \rho)\bar{\sigma}^2 + \rho\sigma_b^2, \tag{3}$$

where $\rho$ is the momentum. On testing, these running statistics are used in place of the batch statistics, ensuring that normalization is independent of the batch size. In this paper, we denote the running statistics $\{\bar{\mu}, \bar{\sigma}^2\}$ of all BN layers by $\bar{S}$, and the batch statistics $\{\mu_b, \sigma_b^2\}$ of all BN layers by $S_b$. with layer indices omitted for simplicity.

## 2.2 FEDERATED LEARNING

FL aims to learn a global model from multiple clients' data while preserving privacy. Let the private local dataset of client $k$ be $D_k = \{(x_i, y_i)\}_{i=1}^{N_k}$, where $x_i$ is the feature of the $i$th sample, $y_i$ is the corresponding class label, and $N_k$ is the total number of samples in $D_k$. These samples are drawn i.i.d. from some joint distribution $P_k(x, y)$. Assume that there are $K$ clients and a server is in charge of communication. The objective of federated learning is to learn a global model $f$ (with parameter set $\{w^g, \bar{S}^g\}$) that minimizes the weighted sum of local empirical losses:

$$\min_{w^g, \bar{S}^g} \sum_{k=1}^{K} \frac{N_k}{N} \mathbb{E}_{(x_i, y_i) \sim P_k} \ell(f(x_i; w^g, \bar{S}^g), y_i), \tag{4}$$

where $f(x_i; \cdot, \cdot)$ is the output vector of predicted class probabilities, $N = \sum_{k=1}^{K} N_k$, and $\ell$ is the loss function. We denote by $\{w^k, \bar{S}^k\}$ the weights and running statistics maintained by client $k$.

The global model weight $w^g$ is obtained by aggregating the local weights $w^k$, while the global statistics $\bar{S}^g$ are obtained by aggregating the local running statistics $\bar{S}^k$. During local training, client $k$ computes the batch BN statistics $S_b^k$ for a batch $b$. Federated Averaging (FedAvg) (McMahan et al., 2017) (Algorithm 1 in Appendix B) optimizes the objective in (4) by iteratively distributing the current global model $\{w^g, \bar{S}^g\}$ to clients, updating local models $\{w^k, \bar{S}^k\}$ on their private data, then averaging these updates with weights proportional to each client's data size $N_k$ to form the next global model.

Subsequent research address challenges of data heterogeneity across clients, also known as non-IID FL (Li et al., 2020; Hsu et al., 2019; Zhao et al., 2018). This refers to varying data distributions across clients, i.e., $P_i(x, y) \neq P_j(x, y)$ for any two clients $i$ and $j$. Some studies tackle client drifts via regularization, which helps prevent local models from deviating too much (Li et al., 2020; Acar et al., 2021). Other approaches share a similar objective of mitigating client drift but achieve it through the introduction of drift variables and variance reduction (Karimireddy et al., 2020; Gao et al., 2022), or by contrastive learning between model representations (Li et al., 2021a). Another line of work exploits class prototypes to guide local training (Tan et al., 2022; Xu et al., 2023).

Li et al. (2021b) define **feature-shift FL** as the setting that includes covariate shift, where the marginal feature distributions across clients differ (i.e., $P_i(x) \neq P_j(x)$), and concept shift, where the conditional distributions vary (i.e., $P_i(x|y) \neq P_j(x|y)$) while the label distribution $P(y)$ remains the same across clients. In practice, most commonly used benchmarks (e.g., *PACS* (Li et al., 2017) and *DomainNet* (Peng et al., 2019)) primarily manifest covariate shift, as the domains differ in feature distributions while $P(y|x)$ is assumed to remain unchanged. In this paper, we also focus on the covariate shift setting. Recent works tackle this feature shift problem through different strategies: prototype-based methods such as FedPlvm (Wang et al., 2024), representation-level augmentation such as FedFA (Zhou & Konukoglu, 2023) and FRaug (Chen et al., 2023), and data-level augmentation such as FedRDN (Yan et al., 2025).

## 2.3 BATCH NORMALIZATION IN FEDERATED LEARNING

Recent works (Hsieh et al., 2020; Wang et al., 2023; Du et al., 2022; Guerraoui et al., 2024; Chen et al., 2025) observe that BN may not work well in FL with heterogeneous data. Early at-

tempts (Hsieh et al., 2020; Du et al., 2022) address this issue by replacing BN with alternative normalization, such as Group Normalization (Wu & He, 2018) and Layer Normalization (Ba et al., 2016). However, they may sacrifice some key benefits of BN, such as faster convergence (Ioffe & Szegedy, 2015) and better generalization (Luo et al., 2018; Bjorck et al., 2018). Moreover, they require extra computation during inference. FedTAN (Wang et al., 2023) demonstrates that performance degradation arises from the mismatch between local and global BN statistics. It proposes aggregating and distributing the global statistics layer by layer during local training. However, this leads to impractical training latency and communication overhead. FBN (Guerraoui et al., 2024) modifies the aggregation of BN statistics to obtain unbiased global estimates and uses them to normalize activations during local training. PN (Wang et al., 2025) replaces batch statistics with population-level mean and variance learned as trainable parameters, thereby alleviating the instability caused by small batch sizes.

Most of the above methods focus on non-IID labels, neglecting the challenges posed by feature shifts across clients in FL. FedBN (Li et al., 2021b) and SiloBN (Andreux et al., 2020) address feature heterogeneity by keeping BN layers or statistics local. However, both approaches target cross-silo FL and rely on stateful clients, making them less practical for cross-device FL or for scenarios with unseen clients where a single global model is required. FedWon (Zhuang & Lyu, 2024) instead tackles feature heterogeneity by removing all BNs and employing weight standardization, but it is limited to CNNs and sacrifices compatibility with widely available pretrained backbones, reducing both convergence and training stability.

# 3 PROPOSED METHODOLOGY

The original design of BN, while effective in centralized training, is ill-suited for feature-shift FL. Section 3.1 first explains the challenge of BN in FL under non-IID features. Section 3.2 introduces a simple and general method that retains BN layers while still enabling a shared global model, unlike prior approaches that either personalize or discard BN. The recent work HBN (Chen et al., 2025) also shares this goal, but achieves it by decoupling the updates of BN statistics and weight parameters. Although effective in the setting with non-IID labels, its performance degrades under feature-shift FL as will be seen in Section 4. By default in FL, a sample $x$ is predicted using $w^g$ together with $\bar{S}^g$. To make the distinction explicit, we let $S_b^g$ denote the batch statistics computed on a test batch $b$ by the global model.

## 3.1 TRAINING-TEST INCONSISTENCY OF BN IN FEATURE-SHIFT FL

In centralized training, data are drawn from a single distribution, so the running statistics $\bar{S}$ can reliably approximate the population mean and variance. As a result, switching from batch statistics $S_b$ during training to running statistics $\bar{S}$ at test time does not compromise performance. On the other hand, in FL with non-IID features, while each client's local statistics $\bar{S}^k$ are reliable estimators for their respective local data distribution, averaging them to form the global statistics $\bar{S}^g$ produces quantities that do not accurately represent any client. As shown in Figure 1 (Section 1), the variance of BN statistics across clients is large in feature-shift FL, meaning that $\bar{S}^g$ deviates substantially from $\bar{S}^k$ on average.

This discrepancy becomes critical at inference: activations are normalized using $\bar{S}^g$, so the BN outputs follow a distribution different from that seen during local training with $S_b^k$. The mismatch between training and testing normalizers introduces a distributional drift at every BN layer. Unlike the familiar covariate shift across clients (caused by different input feature distributions), this is an internal shift induced by the BN layers themselves. Modern networks have many BN layers (e.g., 20 in ResNet-18), and each layer introduces a drift. As activations pass through the network, these drifts accumulate with depth, eventually leading to significant errors at the output.

This explains the observation in Section 1: using $\bar{S}^g$ at test time introduces drift and degrades performance, whereas batch statistics better match the local distribution and can improve accuracy. In contrast, in label-shift FL, clients share similar feature distributions, making $\bar{S}^g$ reliable, whereas batch statistics merely add noise and order dependence.

To further illustrate this training-test inconsistency, we define the test-time loss gap $\Delta^{te} = \ell(f(x; w^g, \bar{S}^g), y) - \ell(f(x; w^g, S_b^g), y)$, which measures the loss difference between using batch statistics versus global statistics (which provide more reliable normalization) at testing. Figure 2 shows $\Delta^{te}$ on the *PACS* dataset at each training round (after each communication and aggregation step). As can be seen, the gap for FedAvg is always positive, confirming that BN with $\bar{S}^g$ harms performance compared to using batch statistics. For comparison, we also show the gap corresponding to centralized training (with $\bar{S}^g \to \bar{S}$ and $\bar{S}_b^g \to S_b$) after each training epoch. Its gap values are all negative, indicating that its running statistics outperform batch statistics at test time.

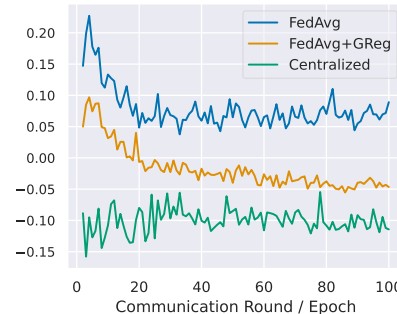

Figure 2: Test-time loss gap $\Delta^{te}$ on *PACS*. Definitions are in Section 3.

### 3.2 LOCAL-GLOBAL CONSISTENCY REGULARIZATION

Our goal is to minimize the test-time loss gap $\Delta^{te}$. However, this is not feasible, as it depends on the updated global statistics $\bar{S}^g$, which are available only after aggregation at the end of each round. Thus, in the following, we consider the analogous training-time loss gap $\Delta^{tr} = \ell(f(x; w^k, \bar{S}^g), y) - \ell(f(x; w^k, S_b^k), y)$. As shown in Figure 3, for centralized training, $\Delta^{tr}$ starts from a negative value and gradually converges to zero. This is because the running statistics $\bar{S}$ (initialized from the pretrained model) are more reliable than the noisy per-batch statistics in early epochs, as the batch contains samples from multiple domains. As training progresses, the activations stabilize and $\bar{S}$ converges toward the population statistics, shrinking the gap to zero. In contrast, $\Delta^{tr}$ for FedAvg is

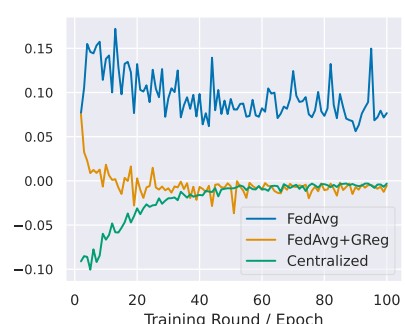

Figure 3: Training-time loss gap $\Delta^{tr}$ on *PACS*. Definitions are in Section 3.

always positive. Although it is also initialized from the pretrained model, $\bar{S}^g$ deviate substantially from each client's local $\bar{S}^k$ under non-IID features (Section 3.1). Thus, replacing $S_b^k$ with $\bar{S}^g$ increases the loss.

This analysis highlights the role of aligning model weights and running statistics. In centralized training, the convergence of $\Delta^{tr}$ to zero indicates that the model becomes agnostic to whether batch or running statistics are used. This invariance is desirable because it helps local training performance translate more reliably to test time. Our aim is therefore to encourage a similar zero-converging training-time loss gap as observed in centralized training.

To close this gap, we propose Local–Global Consistency Regularization (GReg), which encourages the model to produce consistent final outputs when BN is applied with either the local batch statistics $S_b^k$ or global statistics $\bar{S}^g$. Concretely, we penalize the divergence between the two output distributions using the symmetrized KL divergence between $q_b$ and $q_g$, which is defined as:

$$\ell_{\text{reg}} = \tfrac{1}{2}\text{KL}(q_b \| q_g) + \tfrac{1}{2}\text{KL}(q_g \| q_b), \tag{5}$$

where $q_b = f(x; w^k, S_b^k)$ (resp. $q_g = f(x; w^k, \bar{S}^g)$) is the output probability vector when BN uses the batch statistics $S_b^k$ (resp. global statistics $\bar{S}^g$). Hence, starting from the second training round (after obtaining the first $\bar{S}^g$ via aggregation), the local model $w^k$ minimizes the total loss

$$\mathcal{L} = \ell(f(x; w^k, S_b^k), y) + \alpha \ell_{\text{reg}}, \tag{6}$$

where $\alpha$ is a tradeoff parameter. Figure 3 shows that with GReg, the training-time loss gap converges to zero, resembling the behavior of centralized training.

To further justify the regularizer $\ell_{\text{reg}}$, consider the cross-entropy loss. $\ell(q, y) = -\log q^y$, where $q^y$ is the predicted probability assigned to the ground-truth class $y$. The following Proposition shows that $\ell_{\text{reg}}$ provides an upper bound on the absolute loss gap. Proof is in Appendix C.

**Proposition 1** (Loss Gap). *For any sample* $(x, y)$, *we have* $\left| \ell(q_g, y) - \ell(q_b, y) \right| \le \frac{2}{\tau} \sqrt{\ell_{\text{reg}}}$, *where* $\tau = \min\{q_b^y, q_g^y\}$.

Hence, minimizing $\ell_{\text{reg}}$ drives the training-time loss gap $\Delta^{tr}$ to zero as in Figure 3. Consequently, from Figure 2 (orange curve), $\Delta^{te}$ of GReg drops below zero, approximating the negative gap seen in centralized training. This implies that GReg improves training–test consistency, preventing performance degradation when using global statistics. A potential concern is the value of $\tau$ in Proposition 1 might be small, making the bound uninformative. We provide an analysis in Appendix D.

In round $t$, $\Delta^{tr}$ uses the broadcasted $\bar{S}_t^g$, whereas $\Delta^{te}$ (post-aggregation) uses $\bar{S}_{t+1}^g$, thus, introducing a one-round offset. To make $\Delta^{tr}$ a meaningful surrogate for $\Delta^{te}$, we apply an exponential moving average scheme on the server side for $\bar{S}^g$ to stabilize the changes. Concretely, the update after aggregation is defined as $\bar{S}_{t+1}^g \leftarrow (1 - \hat{\rho})\bar{S}_t^g + \hat{\rho}\bar{S}_{t+1}^g$, with momentum parameter $\hat{\rho} = 0.1$ by default. This smoothing prevents sharp changes of $\bar{S}^g$, making GReg more reliable to minimize $\Delta^{te}$. Appendix B presents the full algorithm of GReg applied to FedAvg. Note that GReg can be applied to any other FL algorithm. We also experimented with several intuitive alternatives for utilizing $\bar{S}^g$ during local training. These approaches turned out to be largely ineffective compared to GReg. Detailed discussions are in Appendix E.3.

## 4 EXPERIMENTS

### 4.1 EXPERIMENTAL SETUPS

**Datasets.** We perform extensive experiments on four image classification benchmarks exhibiting non-IID features due to varying domains: (i) *PACS* (Li et al., 2017), with domains: Art Painting, Cartoon, Photo, and Sketch; (ii) *DomainNet* (Peng et al., 2019), with domains: Clipart, Infograph, Painting, Quickdraw, Real, and Sketch (we use the top-10 classes subset of *DomainNet*); (iii) *VLCS* (Ghifary et al., 2015), with domains: Caltech101, LabelMe, SUN09, and VOC2007; and (iv) *Camelyon17* (Koh et al., 2021), which is a histopathology dataset collected from five hospitals. For all datasets, we regard each domain as a local client, which naturally induces feature shifts across clients. More details about the splits can be found in Appendix E.1.

**Baselines.** To evaluate GReg, we focus on methods that aim to learn a shared global model, including both standard FL baselines and FL methods proposed to improve performance under non-IID features. Specifically, we integrate GReg with (i) FedAvg (McMahan et al., 2017); (ii) FedProx (Li et al., 2020); (iii) FedPlvm (Wang et al., 2024); a recent representative of prototype learning; (iv) FedFA (Zhou & Konukoglu, 2023), which is based on representation augmentation approach; and (v) FedRDN (Yan et al., 2025), a data-level augmentation by randomly injecting statistics from other clients into local data. We further compare with three normalization-aware methods that explicitly address BN under non-IID FL: (i) HBN (Chen et al., 2025); (ii) FedWon (Zhuang & Lyu, 2024); and (iii) FBN (Guerraoui et al., 2024). For reference, we also report centralized training, which serves as a performance upper bound.

**Implementation Details.** We use ResNet-18 (He et al., 2016), pretrained on the ImageNet (Deng et al., 2009), as backbone, followed by a linear classifier. Training is conducted for 100 communication rounds, with one local epoch per round. We use SGD with a batch size of 16 and a default learning rate of 0.01. For *VLCS*, the learning rate is set to 0.005 to ensure training stability. For hyperparameters, we set the proximal term coefficient in FedProx to $\mu = 10^{-3}$ and the loss term coefficient in FedPlvm to $\lambda = 1$, while the other parameters follow the defaults in their respective papers. For GReg, we use the default choices of $\alpha = 1$ and $\hat{\rho} = 0.1$ to avoid hyperparameter tuning.

### 4.2 MAIN RESULTS

Table 2 reports the average test accuracy across clients of centralized training and FL methods. The full per-client results are provided in Appendix E.2 (Tables 7 and 8). Overall, the proposed GReg consistently improves on almost FL methods across different datasets, with the only exception being FedFA, where the gains are smaller on milder benchmarks. For example, on *PACS*, GReg improves FedAvg by 3.5%, and achieves the highest accuracy when combined with FedFA, reaching within 0.2% of centralized training. GReg still provides consistent improvements on the milder feature-

Table 2: Average test accuracies (%) across clients of centralized training and FL methods on feature-shift benchmarks. **Bold** indicates performance improvement with GReg. All results are averaged over three runs.

| Method | PACS | DomainNet | VLCS | Camelyon17 | Average |
|---|---|---|---|---|---|
| Central | 94.0 | 85.7 | 81.2 | 96.4 | 89.3 |
| FedAvg | 89.8 | 81.3 | 79.9 | 94.5 | 86.4 |
| +GReg | **93.3** | **84.2** | **80.3** | **95.9** | **88.4** |
| FedProx | 89.5 | 81.5 | 80.2 | 94.5 | 86.4 |
| +GReg | **93.3** | **83.9** | **80.7** | **96.0** | **88.5** |
| FedPlvm | 88.8 | 82.1 | 80.0 | 94.5 | 86.4 |
| +GReg | **93.0** | **84.3** | **81.2** | **96.3** | **88.7** |
| FedFA | 92.4 | 83.3 | 81.0 | 95.4 | 88.0 |
| +GReg | **93.8** | **84.3** | 80.4 | 95.0 | **88.4** |
| FedRDN | 90.5 | 81.2 | 79.7 | 94.8 | 86.6 |
| +GReg | **92.9** | **83.8** | **80.8** | **96.0** | **88.4** |
| HBN | 90.7 | 82.2 | 80.1 | 94.7 | 86.9 |
| FedWon | 91.4 | 76.9 | 74.2 | 95.3 | 84.5 |
| FBN | 79.8 | 63.0 | 70.3 | 93.2 | 76.6 |

shift benchmarks of *VLCS* and *Camelyon17*, [3] though the improvement is limited as FedFA already augments feature representations extensively.

Another important takeaway from Table 2 is that simply applying GReg to FedAvg already achieves accuracies comparable to state-of-the-art methods for feature-shift FL. This motivates us to rethink the underlying challenge in feature-shift FL, suggesting that the primary bottleneck may lie in the mismatch of BN statistics, which emerges as a major source of performance degradation. Unlike normalization-aware methods such as HBN, FedWon, and FBN, GReg achieves superior performance without modifying the architecture or BN layers, making it a simple plug-and-play solution.

### 4.3 SENSITIVITIES TO HYPERPARAMETERS

In this section, we examine the sensitivity of GReg on *PACS*, with FedAvg serving as the baseline, while keeping other settings fixed.

**Sensitivity to $\alpha$ in Eq.(6)** In this experiment, we evaluate GReg by varying $\alpha$, the tradeoff coefficient for $\ell_{\text{reg}}$. As can be seen from Figure 4, a moderate value of $\alpha \in [1.5, 2.0]$ yields the best performance. When $\alpha$ is too small, the regularization is underweighted and the training–test inconsistency is insufficiently corrected Conversely, overly large values make the regularization dominate the objective and hinder convergence. Overall, GReg remains competitive with FedAvg across a wide range of $\alpha$.

**Sensitivity to Batch Size** Figure 5 shows the effect of varying the batch size. As can be seen, GReg is largely insensitive and consistently outperforms FedAvg. Moreover, interestingly, in feature-shift FL, using a larger batch does not necessarily improve performance. We speculate that a large batch size amplifies the bias toward each client's local domain while reducing the beneficial stochasticity of smaller batches (Keskar et al., 2016).

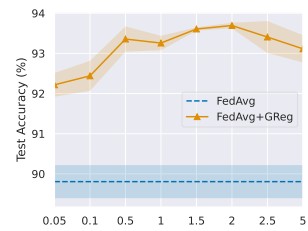

Figure 4: Sensitivity to $\alpha$ on *PACS*.

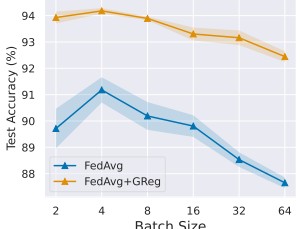

Figure 5: Sensitivity to batch size on *PACS*.

[3]For *VLCS* and *Camelyon17*, the distributional differences mainly arise from collection biases across sub-datasets rather than drastic style variations as in *PACS* and *DomainNet*. This is evident since domains in *VLCS* are all natural photographs, and domains in *Camelyon17* are all medical slides.

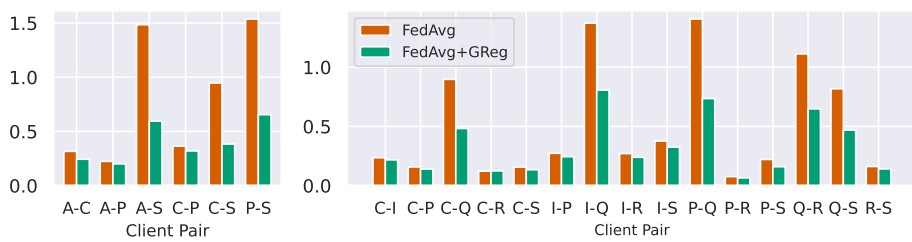

Figure 6: Representation divergence between clients on *PACS* (Left), and *DomainNet* (Right).

## 4.4 Representation Distribution Divergence Among Clients

We study the discrepancies in representation distributions between client domains. Specifically, we examine the penultimate-layer outputs on each client's test set. Assume that every channel follows a normal distribution $N(\cdot, \cdot)$. We first measure the divergence between client domains $i, j$ for each channel: $D(N_i \| N_j) = KL(N_i \| N_j) + KL(N_j \| N_i)$, and then average over all channels to obtain the representation divergence between the two domains. This metric has been similarly used in domain adaptation (Li et al., 2016).

Figure 6 shows the representation divergence between clients. As observed, GReg exhibits lower divergence than FedAvg, especially between highly dissimilar domains such as Photo (P) vs. Sketch (S) in *PACS*, and Quickdraw (Q) vs. Photo (P) in *DomainNet*. This demonstrates that GReg's use of global statistics encourages more domain-invariant representations, thereby mitigating domain shifts across clients.

Table 3: Test accuracies (%) of FL methods on the held-out unseen clients on *PACS* and *DomainNet*.

| Method | *PACS* | | | | | *DomainNet* | | | | | | |
|--------|------|------|------|------|------|------|------|------|------|------|------|------|
| | A | C | P | S | Avg. | C | I | P | Q | R | S | Avg. |
| FedAvg | 75.0 | 63.5 | 94.2 | 57.6 | 72.6 | 85.9 | 40.3 | 85.3 | 52.6 | 84.8 | 82.0 | 71.8 |
| +GReg | 82.3 | 76.1 | 95.4 | 66.7 | **80.1** | 88.8 | 41.2 | 87.9 | 53.6 | 87.3 | 85.2 | **74.0** |
| FedProx | 77.1 | 63.6 | 92.2 | 57.7 | 72.7 | 87.3 | 40.2 | 83.8 | 52.0 | 84.5 | 81.6 | 71.6 |
| +GReg | 83.1 | 73.9 | 95.6 | 68.0 | **80.1** | 89.2 | 42.0 | 87.2 | 54.7 | 85.7 | 84.8 | **73.9** |
| FedPlvm | 77.6 | 60.8 | 91.6 | 61.6 | 72.9 | 87.8 | 44.3 | 85.6 | 51.2 | 85.0 | 81.2 | 72.5 |
| +GReg | 81.6 | 75.6 | 95.2 | 71.2 | **80.9** | 87.8 | 44.9 | 88.7 | 53.7 | 86.4 | 85.6 | **74.5** |
| FedFA | 84.1 | 71.0 | 95.4 | 59.8 | 77.6 | 89.3 | 44.0 | 89.3 | 52.2 | 84.9 | 86.3 | 74.3 |
| +GReg | 86.0 | 74.0 | 96.0 | 71.2 | **81.8** | 90.1 | 42.2 | 88.8 | 55.0 | 86.6 | 87.4 | **75.0** |
| HBN | 72.2 | 66.6 | 93.0 | 62.3 | 73.5 | 85.2 | 41.9 | 85.0 | 51.3 | 84.5 | 81.4 | 71.5 |
| FedWon | 68.8 | 71.0 | 80.8 | 67.0 | 71.9 | 81.6 | 38.7 | 81.7 | 52.5 | 80.3 | 80.7 | 69.2 |

## 4.5 Generalization to Unseen Clients

After training the global model, deploying it to new clients is substantially more difficult, as their domains may be entirely unseen and can even differ from all training domains. This makes the problem more challenging than serving clients within the federation. In this experiment, we examine whether GReg helps tackle test-time feature shift. Similar to (Zhou & Konukoglu, 2023), we use the leave-one-client-out protocol, i.e., each time select one client as the unseen test client and train on the remaining $K - 1$ clients. Table 3 shows the accuracies on the unseen clients. We observe that applying GReg consistently improves performance across almost all domains (except on the Infograph and Painting domains when applied to FedFA). In particular, when applied to FedAvg, it yields an average gain of $7.5\%$ on *PACS* and $2.2\%$ on *DomainNet*. FedFA serves as a strong baseline in this scenario, as it utilizes comprehensive representation-level augmentation for regularization. With GReg applied on top, its performance is further boosted to $81.8\%$ on *PACS* and $75.0\%$ on *DomainNet*, achieving the highest results. In contrast, HBN provides only marginal improvements over FedAvg, indicating that its asynchronous treatment of weights and statistics does not translate

Table 4: Test accuracies (%) of different network architectures on the four feature shift datasets.

| Architecture | #Params | PACS | | DomainNet | | VLCS | | Camelyon17 | |
|---|---|---|---|---|---|---|---|---|---|
| | | FedAvg | +GReg | FedAvg | +GReg | FedAvg | +GReg | FedAvg | +GReg |
| ResNet-50 | 23.5M | 92.0 | **94.8** | 85.1 | **86.9** | 80.6 | **81.9** | 92.9 | **96.1** |
| EfficientNet-B0 | 4.0M | 81.9 | **91.9** | 77.0 | **81.7** | 71.6 | **74.9** | 87.6 | **94.1** |
| MobileNetV2 | 2.2M | 89.1 | **93.1** | 81.7 | **84.3** | 79.6 | **80.5** | 94.6 | **96.4** |
| DenseNet-121 | 7.0M | 92.6 | **95.6** | 84.5 | **87.4** | 80.7 | **81.8** | 94.7 | **96.3** |
| LeViT-128s | 7.0M | 89.4 | **94.0** | 82.4 | **85.3** | 78.7 | **80.6** | 95.5 | **96.7** |
| MobileViT-s | 4.9M | 89.5 | **94.6** | 82.0 | **86.0** | 82.4 | **83.1** | 94.1 | **96.5** |

Table 5: Test accuracies (%) of cross-device FL with 100 clients on *PACS* and 200 clients on *DomainNet*. Training is run for 100 rounds on *PACS* and 400 rounds on *DomainNet*.

| Method | PACS | | | | | DomainNet | | | | | | |
|---|---|---|---|---|---|---|---|---|---|---|---|---|
| | A | C | P | S | Avg. | C | I | P | Q | R | S | Avg. |
| FedAvg | 83.1 | 81.5 | 96.2 | 71.9 | 83.2 | 74.8 | 26.4 | 67.1 | 36.1 | 80.1 | 59.4 | 57.3 |
| +GReg | 87.6 | 86.9 | 97.2 | 80.2 | **88.0** | 74.9 | 27.8 | 66.5 | 55.2 | 79.1 | 59.9 | **60.6** |
| HBN | 82.8 | 83.5 | 96.6 | 74.3 | 84.3 | 71.3 | 24.6 | 63.8 | 36.4 | 77.6 | 57.1 | 55.1 |

well to new clients. FedWon performs even worse, often outperformed by FedAvg, suggesting that sacrificing BN is detrimental for unseen-domain generalization.

## 4.6 Use of Other Network Architectures

In this experiment, we experiment with different network architectures, including CNNs (ResNet-50, EfficientNet-B0, MobileNetV2, DenseNet-121) and efficient ViT variants (LeViT-128s, MobileViT-s) that are much smaller than vanilla ViTs (Graham et al., 2021; Mehta & Rastegari, 2022) and thus better suited for FL. More details are provided in Appendix E.5. As can be seen from Table 4, GReg consistently improves accuracy over FedAvg. These highlight that GReg is effective across BN-equipped networks.

## 4.7 Cross-device FL with Partial Participation

In this experiment, we study the cross-device FL setting on *PACS* and *DomainNet*, which involves a large number of stateless clients with unreliable participation across training rounds. Table 5 reports the test accuracies of FedAvg, FedAvg+GReg, and HBN under partial participation, where 10% of clients are randomly selected each round. On *PACS*, we partition the dataset into 100 clients, each with 46 training samples, while on *DomainNet*, we use the Top-100 classes subset to construct 200 clients, each with 380 training samples.

As shown, GReg consistently improves upon FedAvg, raising the average accuracy by 4.8% on *PACS* and 3.3% on *DomainNet*. In contrast, HBN provides only marginal gains on *PACS* and degrades on *DomainNet*, where the federation involves more clients and larger local datasets. GReg is simple and inherently suitable for cross-device FL, as it does not require clients to maintain any parameters. Clients can seamlessly join training upon receiving the global model parameters $\{w^g, \bar{S}^g\}$.

## 5 Conclusion

In this paper, we studied the performance degradation of BN in feature-shift FL. Our analysis identifies the root cause as the training–test inconsistency stemming from the large variation in local BN running statistics across clients. To address this issue, we proposed GReg, a simple regularization approach that explicitly aligns the predictions obtained under batch statistics and global statistics during local training. GReg requires no architectural modifications and incurs no additional communication cost. Extensive experiments on four benchmarks demonstrate that GReg consistently improves the performance of most FL methods in feature-shift FL. Its simplicity and versatility make GReg a promising solution for real-world FL applications involving non-IID features.

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

## A EXPERIMENTAL SETUP FOR *PACS* AND LABEL NON-IID *CIFAR-10* COMPARISON

In Section 1, we presented an illustrative experiment contrasting feature-shift FL (*PACS*) with label-shift FL (*CIFAR-10*). Here we describe the setup in detail. For *PACS*, we follow the common protocol and treat each domain (Photo, Art painting, Cartoon, Sketch) as a separate client, resulting in four clients in total. We use the train/test splits summarized in Table 6. For *CIFAR-10*, we construct a label non-IID setting with 10 clients by subsampling and partitioning the training data according to a Dirichlet distribution with concentration parameter $\alpha = 0.3$ over the class labels. Each client thus receives a distinct but imbalanced class distribution, mimicking realistic label skew, with 2,500 training samples per client. The test set remains unchanged from the standard split, containing 10,000 samples.

All clients use a ResNet-18 backbone followed by a classifier layer trained with FedAvg for 100 rounds. We adopt SGD with a learning rate of $0.01$ and perform one local epoch per round for both datasets. During evaluation, we compare two inference modes for BN layers: (i) using the global BN statistics aggregated throughout federated training, and (ii) using the test-time batch statistics. This allows us to isolate the effect of BN statistic mismatch under feature- vs. label-shift scenarios.

## B ALGORITHMS

Algorithm shows the standard Federated Averaging (FedAvg) algorithm, while Algorithm 2 shows the algorithm when GReg is integrated with FedAvg.

---

**Algorithm 1:** FedAvg.

---

**Input:** Client datasets $\{D_k\}_{k=1}^{K}$, communication rounds $T$, learning rate $\eta$
**Initialize:** Global weights $w^g$; global BN statistics $\bar{S}^g$
**Output:** Final global model $\{w_T^g, \bar{S}_T^g\}$
**for** $t = 0, 1, \ldots, T-1$ **do**
    // Broadcast
    Server sends $\{w_t^g, \bar{S}_t^g\}$ to all clients
    // Client step
    **For each** *client* $k$ **do**
        Initialize local model $\{w^k, \bar{S}^k\}$ with $\{w_t^g, \bar{S}_t^g\}$
        **For each** *batch* $\{(x, y)\} \subset D_k$ **do**
            Compute the cross-entropy loss $\ell$ on each sample $(x, y)$
            Update $w^k \leftarrow w^k - \eta \nabla_{w^k} \ell$
            Update running stats $\bar{S}^k$ via (2) and (3)
        Client uploads $\{w^k, \bar{S}^k\}$ to server
    // Server aggregation
    $w_{t+1}^g \leftarrow \sum_{k=1}^{K} \frac{N_k}{N} w^k$
    $\bar{S}_{t+1}^g \leftarrow \sum_{k=1}^{K} \frac{N_k}{N} \bar{S}^k$
**return** $\{w_T^g, \bar{S}_T^g\}$

---

## C PROOF OF PROPOSITION 1

**Proposition** (Loss Gap). *For any sample* $(x, y)$*, let* $\tau = \min\{q_b^y, q_g^y\}$*. Then the following holds:*

$$\left| \ell(q_g, y) - \ell(q_b, y) \right| \leq \frac{2}{\tau} \sqrt{\ell_{\text{reg}}}.$$

*Proof.* Consider a sample $(x, y)$. We denote by $q_g^c$ and $q_b^c$ the predicted probabilities of class $c$ under global and batch statistics, respectively. Recall that $\ell_{\text{reg}} = \frac{1}{2}\text{KL}(q_b \| q_g) + \frac{1}{2}\text{KL}(q_g \| q_b)$,

---

**Algorithm 2:** FedAvg + GReg.

---

**Input:** Client datasets $\{D_k\}_{k=1}^K$, communication rounds $T$, learning rate $\eta$, loss tradeoff
    parameter $\alpha$, server side momentum $\hat{\rho}$ for BN statistics
**Initialize:** Global weights $w^g$; global BN statistics $\bar{S}^g$
**Output:** Final global model $\{w_T^g, \bar{S}_T^g\}$
**for** $t = 0, 1, \ldots, T-1$ **do**
  // Broadcast
  Server sends $\{w_t^g, \bar{S}_t^g\}$ to all clients
  // Client step
  **For each** *client k* **do**
    Initialize local model $\{w^k, \bar{S}^k\}$ with $\{w_t^g, \bar{S}_t^g\}$
    **For each** *batch* $\{(x,y)\} \subset D_k$ **do**
      **if** $t = 0$ **then**
        $\mathcal{L} \leftarrow \ell\big(f(x; w^k, S_b^k), y\big)$
      **else**
        Compute $q_b \leftarrow f(x; w^k, S_b^k)$ and $q_g \leftarrow f(x; w^k, \bar{S}_t^g)$
        $\ell_{\text{reg}} \leftarrow \frac{1}{2}\text{KL}(q_b \| q_g) + \frac{1}{2}\text{KL}(q_g \| q_b)$  (Eq. 5)
        $\mathcal{L} \leftarrow \ell\big(f(x; w^k, S_b^k), y\big) + \alpha\, \ell_{\text{reg}}$  (Eq. 6)
      Update $w^k \leftarrow w^k - \eta \nabla_{w^k} \mathcal{L}$
      Update running stats $\bar{S}^k$ via (2) and (3)
    Client uploads $\{w^k, \bar{S}^k\}$ to server
  // Server aggregation
  $w_{t+1}^g \leftarrow \sum_{k=1}^K \frac{N_k}{N} w^k$
  $\bar{S}_{t+1}^g \leftarrow \sum_{k=1}^K \frac{N_k}{N} \bar{S}^k$
  // Momentum update for BN statistics stability
  $\bar{S}_{t+1}^g \leftarrow (1 - \hat{\rho})\, \bar{S}_t^g + \hat{\rho}\bar{S}_{t+1}^g$
**return** $\{w_T^g, \bar{S}_T^g\}$

---

$$\big|\ell(q_g, y) - \ell(q_b, y)\big| = \big| -\log q_g^y + \log q_b^y \big| \tag{7}$$

$$\leq \frac{\big|q_b^y - q_g^y\big|}{\min\{q_b^y, q_g^y\}} \tag{8}$$

$$= \frac{1}{\tau}\big|q_b^y - q_g^y\big| \tag{9}$$

$$\leq \frac{1}{\tau}\sum_c \big|q_b^c - q_g^c\big| \tag{10}$$

$$= \frac{1}{\tau}\|q_b - q_g\|_1 \tag{11}$$

$$\leq \frac{1}{\tau}\sqrt{2\text{KL}(q_b\|q_g)} \tag{12}$$

$$\leq \frac{2}{\tau}\sqrt{\ell_{\text{reg}}} \tag{13}$$

Inequality (8) simply uses the fact that $\big|\log a - \log b\big| \leq \frac{|a-b|}{\min\{a,b\}}$, which can be easily proved by using the Mean Value Theorem. Inequality (12) uses Pinsker's inequality. □

## D ANALYSIS ON $\tau$ IN PROPOSITION 1

A potential concern is the value of $\tau$ in Proposition 1. If it is too tiny, the bound can be numerically large and uninformative. We provide empirical evidence in Figure 7, which plots the minimum

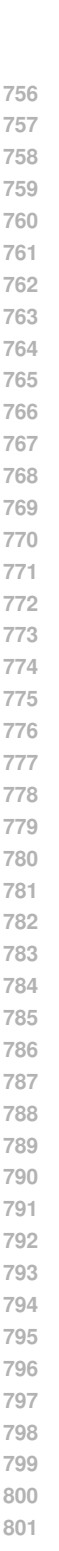

Figure 7: minimum $q_b^y$ and $q_g^y$ over samples on *PACS*.

values of $q_g^y$ and $q_b^y$ across all samples in each round. Both quantities steadily increase as training progresses, implying that $\frac{1}{\tau}$ decreases over time. This trend confirms that the bound remains meaningful throughout training, since the predicted ground-truth probabilities under both batch and global statistics do not collapse toward zero but instead become more confident as the model learns.

# E   EXPERIMENTS

## E.1   DATASET DETAILS

We follow prior work (Li et al., 2021b; Wang et al., 2024) and use limited local training data per client to reflect realistic FL scenarios. For all datasets, we adopt the same preprocessing and transforms as FedBN (Li et al., 2021b). Training/validation splits are balanced across clients, while test sets retain their natural size ratios.

Table 6: Dataset sizes per domain/hospital (train/val/test). Each domain corresponds to one client in our main experiment.

| Dataset | Domain/Hospital | Train | Val | Test |
|---|---|---|---|---|
| DomainNet | Clipart | 630 | 210 | 526 |
| | Infograph | 630 | 210 | 657 |
| | Painting | 630 | 210 | 619 |
| | Quickdraw | 630 | 210 | 1000 |
| | Real | 630 | 210 | 1217 |
| | Sketch | 630 | 210 | 554 |
| PACS | Photo | 584 | 233 | 615 |
| | Art Painting | 584 | 233 | 704 |
| | Cartoon | 584 | 233 | 501 |
| | Sketch | 584 | 233 | 1179 |
| VLCS | Caltech101 | 297 | 297 | 425 |
| | LabelMe | 297 | 297 | 796 |
| | SUN09 | 297 | 297 | 985 |
| | VOC2007 | 297 | 297 | 1013 |
| Camelyon17 | H1 | 293 | 293 | 536 |
| | H2 | 293 | 293 | 315 |
| | H3 | 293 | 293 | 766 |
| | H4 | 293 | 293 | 1169 |
| | H5 | 293 | 293 | 1321 |

**DomainNet** (Peng et al., 2019) is a large-scale dataset with 586,575 images from 345 classes. It contains six domains (Clipart, Infograph, Painting, Quickdraw, Real, Sketch). We follow Li et al.

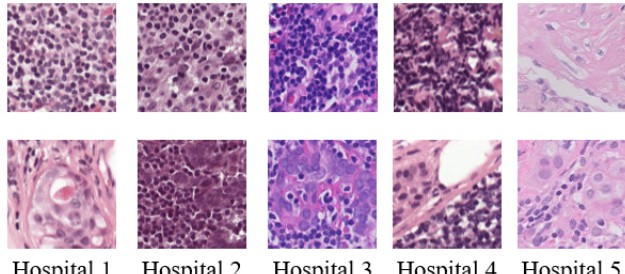

Hospital 1 Hospital 2 Hospital 3 Hospital 4 Hospital 5

Figure 8: Examples of tumor histopathology patches from *Camelyon17*. Top: tumor; Bottom: non-tumor.

Table 7: Test accuracies (%) of FL methods on *PACS* and *DomainNet*. **Bold** indicates performance improvement with GReg. For *PACS*, the abbreviations A, C, P, and S represent Art Painting, Cartoon, Photo, and Sketch. For *DomainNet*, C, I, P, Q, R, and S represent Clipart, Infograph, Painting, Quickdraw, Real, and Sketch. Results are averaged over three runs.

| Method | PACS | | | | | DomainNet | | | | | | |
|---|---|---|---|---|---|---|---|---|---|---|---|---|
| | A | C | P | S | Avg. | C | I | P | Q | R | S | Avg. |
| Central | 91.7 | 93.4 | 98.2 | 92.7 | 94.0 | 92.5 | 56.4 | 92.9 | 86.1 | 93.3 | 92.6 | 85.7 |
| FedAvg | 89.2 | 90.0 | 97.2 | 82.9 | 89.8 | 91.7 | 49.5 | 90.8 | 75.3 | 91.4 | 89.3 | 81.3 |
| +GReg | 90.3 | 92.8 | 98.1 | 92.0 | **93.3** | 93.2 | 52.4 | 92.8 | 82.3 | 93.6 | 91.0 | **84.2** |
| FedProx | 88.9 | 89.6 | 97.0 | 82.7 | 89.5 | 91.8 | 49.9 | 90.8 | 75.3 | 92.1 | 88.9 | 81.5 |
| +GReg | 90.4 | 92.8 | 98.1 | 91.8 | **93.3** | 92.8 | 52.4 | 92.4 | 81.7 | 93.5 | 90.9 | **83.9** |
| FedPlvm | 88.5 | 87.2 | 96.9 | 82.7 | 88.8 | 92.6 | 52.7 | 91.5 | 74.9 | 92.6 | 88.2 | 82.1 |
| +GReg | 91.1 | 92.5 | 98.1 | 90.4 | **93.0** | 93.0 | 54.8 | 93.1 | 81.0 | 93.7 | 90.1 | **84.3** |
| FedFA | 91.4 | 92.8 | 98.0 | 87.4 | 92.4 | 93.9 | 52.2 | 92.6 | 76.2 | 92.8 | 92.1 | 83.3 |
| +GReg | 91.7 | 93.6 | 98.3 | 91.5 | **93.8** | 94.0 | 51.8 | 93.3 | 81.1 | 93.4 | 92.3 | **84.3** |
| FedRDN | 90.4 | 89.5 | 97.3 | 84.9 | 90.5 | 92.3 | 48.8 | 90.8 | 73.8 | 92.6 | 89.1 | 81.2 |
| +GReg | 89.9 | 92.1 | 98.1 | 91.6 | **92.9** | 93.6 | 51.3 | 93.4 | 80.4 | 93.6 | 90.3 | **83.8** |
| HBN | 89.8 | 91.0 | 96.9 | 85.0 | 90.7 | 92.1 | 50.6 | 91.3 | 77.9 | 91.8 | 89.6 | 82.2 |
| FedWon | 87.8 | 91.5 | 94.3 | 92.2 | 91.4 | 85.0 | 47.1 | 82.9 | 79.1 | 84.0 | 83.5 | 76.9 |
| FBN | 71.6 | 78.0 | 92.5 | 76.9 | 79.8 | 74.4 | 39.6 | 67.6 | 50.5 | 74.3 | 71.5 | 63.0 |

(2021b) to use the top-10 classes subset. In the original splits used by them, each client had only 105 training samples; we expand this to 630 per client, to enhance training stability and to ensure that performance gains are not overly dominated by the pretrained initialization.

**PACS** (Li et al., 2017) contains 9,991 images, and they are from four domains (Photo, Art Painting, Cartoon, Sketch). In our experiment, each domain is split 70/30 into train/test, with subsampling to enforce equal client sizes.

**VLCS** (Ghifary et al., 2015) includes 10,729 images from four domains: Caltech101, LabelMe, SUN09, and VOC2007. We apply the same split and subsampling as for PACS.

**Camelyon17** (Koh et al., 2021) is a large-scale dataset that consists of ~400k histopathology patches from five hospitals (H1–H5), each treated as a client in our experiment. We subsample 3% of the ~400k patches and apply the same split protocol. Figure 8 shows examples of the patches.

### E.2  DETAILED EXPERIMENTAL RESULTS

Table 7 shows the detailed test accuracies of GReg and all baselines on each client and their average on *PACS* and *DomainNet*. Table 8 shows the detailed test accuracies on *VLCS* and *Camelyon17*. From Table 7, we observe that the gains of GReg are most pronounced on clients that differ substantially from the others, such as Sketch (+9.1%) and Quickdraw (+7.0%). Compared to other domains in *PACS* and *DomainNet*, these domains consist of hand-drawn line art, lacking the rich textures and colors of photo-like or painting-like images. As a result, their feature distributions are more distant

Table 8: Test accuracies (%) of FL methods on *VLCS* and *Camelyon17*. For *VLCS*, C, L, S, and V represent Caltech101, LabelMe, SUN09, and VOC2007. For *Camelyon17*, H1–H5 correspond to Hospitals 1–5. Results are averaged over three runs.

| Method | *VLCS* | | | | | *Camelyon17* | | | | | |
|---|---|---|---|---|---|---|---|---|---|---|---|
| | C | L | S | V | Avg. | H1 | H2 | H3 | H4 | H5 | Avg. |
| Central | 99.8 | 71.4 | 73.7 | 79.7 | 81.2 | 96.5 | 96.2 | 95.3 | 96.7 | 97.2 | 96.4 |
| FedAvg | 99.5 | 68.4 | 73.4 | 78.3 | 79.9 | 95.5 | 94.9 | 93.9 | 94.9 | 93.5 | 94.5 |
| +GReg | 99.8 | 69.7 | 72.3 | 79.3 | **80.3** | 96.8 | 95.5 | 95.1 | 95.6 | 96.5 | **95.9** |
| FedProx | 99.6 | 68.9 | 73.6 | 78.6 | 80.2 | 95.8 | 94.7 | 94.0 | 95.0 | 93.2 | 94.5 |
| +GReg | 99.8 | 70.2 | 72.6 | 80.0 | **80.7** | 97.1 | 96.2 | 95.0 | 95.5 | 96.3 | **96.0** |
| FedPlvm | 99.6 | 67.8 | 73.8 | 78.7 | 80.0 | 95.8 | 94.4 | 94.0 | 94.6 | 94.0 | 94.5 |
| +GReg | 99.8 | 70.7 | 74.1 | 80.0 | **81.2** | 97.3 | 96.7 | 95.1 | 95.7 | 96.9 | **96.3** |
| FedFA | 99.9 | 67.4 | 75.5 | 81.1 | 81.0 | 96.7 | 94.4 | 94.3 | 95.0 | 96.4 | 95.4 |
| +GReg | 99.8 | 71.7 | 70.8 | 79.4 | 80.4 | 96.0 | 94.8 | 94.1 | 94.2 | 96.2 | 95.0 |
| FedRDN | 99.8 | 66.5 | 73.6 | 79.1 | 79.7 | 96.2 | 93.8 | 94.3 | 95.6 | 94.1 | 94.8 |
| +GReg | 99.8 | 71.1 | 72.7 | 79.8 | **80.8** | 96.3 | 95.8 | 95.3 | 95.6 | 96.8 | **96.0** |
| HBN | 99.5 | 69.0 | 73.6 | 78.1 | 80.1 | 96.6 | 94.8 | 93.5 | 95.5 | 93.2 | 94.7 |
| FedWon | 97.6 | 66.6 | 66.1 | 66.5 | 74.2 | 96.8 | 95.8 | 94.4 | 94.8 | 94.9 | 95.3 |
| FBN | 93.3 | 66.3 | 57.3 | 64.2 | 70.3 | 94.3 | 90.1 | 94.3 | 92.6 | 94.6 | 93.2 |

from the global average, amplifying the training–test inconsistency and magnifying the benefit of $\ell_{\text{reg}}$.

Table 9: Comparison of different ways to utilize the global BN statistics during local training. Average test accuracies (%) over clients are reported.

| Method | PACS | DomainNet |
|---|---|---|
| FedAvg (baseline) | 89.8 | 81.3 |
| Global stats. only | 75.7 | 63.4 |
| Mix stats. ($\lambda$=0.5) | 90.4 | 82.1 |
| Mix stats. (learn $\lambda$) | 90.5 | 82.1 |
| Ensemble | 90.8 | 81.5 |
| **GReg (Ours)** | **93.3** | **84.2** |

### E.3 EXPERIMENTS ON ALTERNATIVE WAYS TO UTILIZE GLOBAL BN STATISTICS

Recall that our motivation is to align $w^k$ with the global statistics $\bar{S}^g$ during local training to mitigate the training–test inconsistency. To this end, we also examine several intuitive alternatives for utilizing $\bar{S}^g$. We use $\bar{\mu}^g$ and $(\bar{\sigma}^g)^2$ to denote the global mean and running variance, respectively. We describe these alternative methods below.

**(i) Global statistics only.** During local training, each BN layer directly uses the global statistics $\bar{\mu}^g$ and $(\bar{\sigma}^g)^2$ for normalization,

$$BN(x) = \frac{x - \bar{\mu}^g}{\sqrt{(\bar{\sigma}^g)^2 + \epsilon}}\gamma + \beta,$$

while the local running estimates $\bar{\mu}^k$ and $(\bar{\sigma}^k)^2$ continue to update according to Eqs. (2) and (3).

**(ii) Mixing statistics.** Each BN layer interpolates between batch statistics $(\mu_b, \sigma_b^2)$ and the global statistics $\bar{S}^g$:

$$\mu_{\text{mix}} = (1 - \lambda)\mu_b + \lambda\,\bar{\mu}^g,$$
$$\sigma_{\text{mix}}^2 = (1 - \lambda)\sigma_b^2 + \lambda\,(\bar{\sigma}^g)^2.$$

The mixed values $(\mu_{\text{mix}}, \sigma_{\text{mix}}^2)$ are then used for normalization in place of $(\mu_b, \sigma_b^2)$. We test both a fixed $\lambda = 0.5$ and a learnable $\lambda$ optimized with the local loss $\ell$.

**(iii) Ensemble.** Each BN layer normalizes activations twice, once with $(\mu_b, \sigma_b^2)$ and once with $(\bar{\mu}^g, (\bar{\sigma}^g)^2)$, and averages the two results:

$$\frac{1}{2}\left( \frac{x - \mu_b}{\sqrt{\sigma_b^2 + \epsilon}} + \frac{x - \bar{\mu}^g}{\sqrt{(\bar{\sigma}^g)^2 + \epsilon}} \right) \gamma + \beta.$$

For all methods, the use of $\bar{S}^g$ begins only after the first aggregation round, once global statistics have been obtained.

Table 9 summarizes the results with FedAvg. As shown, these alternatives are largely ineffective compared to GReg. Mixing statistics and ensembling only incorporate $\bar{S}^g$ indirectly, which fails to enforce strong alignment between $w^k$ and $\bar{S}^g$. Using global statistics only causes a sharp drop in performance, since $\bar{S}^g$ is fixed during each local update but changes abruptly after aggregation, leading to unstable training and poor convergence.

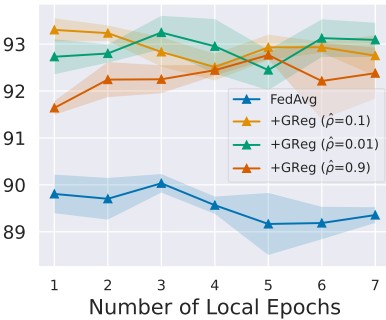

Figure 9: Varying number of local epochs on *PACS*.

### E.4 Effect of Varying Number of Local Epochs

In this experiment, we vary the number of local epochs of GReg and FedAvg on *PACS*, with results also shown for different server-side BN momentum values $\hat{\rho} \in \{0.01, 0.1, 0.9\}$ of GReg. Figure 9 shows that GReg consistently outperforms FedAvg, though a large $\hat{\rho}$ tends to yield slightly lower performance. This validates our design choice to use a small momentum for preventing sharp changes of $\bar{S}^g$.

### E.5 Details of Experiments on Other Network Architectures

Besides the default ResNet-18, we further evaluate GReg on FedAvg using four CNN-based architectures (ResNet-50, EfficientNet-B0, MobileNetV2, and DenseNet-121) and two transformer-based architectures (LeViT-128s and MobileViT-s). The results are shown in Table 4. The transformers are efficient variants of ViT (Dosovitskiy et al., 2020), making them more suitable for FL. In particular, LeViT-128s (Graham et al., 2021) adopts a pyramidal design with downsampling and replaces LayerNorm with BatchNorm after $1 \times 1$ convolutions, enabling faster inference. MobileViT-s (Mehta & Rastegari, 2022) integrates BN-equipped convolutional blocks with lightweight transformer layers to capture both local features and global context. For the CNN architectures, we use the same training hyperparameters as ResNet-18. For LeViT-128s and MobileViT-s, following common practice for training transformers, we adopt AdamW with a learning rate of $1 \times 10^{-4}$ and weight decay of 0.05, while keeping other hyperparameters consistent with the CNN setting.

## F Communication and Training Efficiency

In this section, we evaluate the efficiency of GReg in terms of both communication and training time. On the communication side, GReg incurs no additional cost compared to the simplest baseline, FedAvg, since it does not transmit any extra parameters or statistics beyond the standard model

Table 10: Training time comparison on *PACS* with ResNet-18 (batch size = 16).

| Method | One Local Epoch Time (sec.) | Relative Time |
|---|---|---|
| FedAvg | $1.5428 \pm 0.1056$ | $1\times$ |
| HBN | $2.9696 \pm 0.1527$ | $1.92\times$ |
| FedWon | $2.7032 \pm 0.1543$ | $1.75\times$ |
| FedAvg+GReg | $2.2364 \pm 0.1214$ | $1.45\times$ |

updates. In contrast, HBN can incur substantially higher communication costs when model evaluation is performed throughout training (e.g., for model selection). Because HBN updates $w^k$ and $\bar{S}^k$ asynchronously, $\bar{S}^k$ is always one round behind $w^k$. As a result, each evaluation of the global model requires an additional round of communication to synchronize statistics, effectively doubling the total communication cost if the model is tested every round. Other baselines, such as FedFA (extra representation-level statistics in the FFALayer), FedRDN (data-level normalization statistics), and FedPlvm (class prototypes), also introduce additional communication, but the amount is relatively minor and remains negligible in practice.

On the training side, the overhead of GReg comes from computing the regularization term, which requires an additional forward pass with global statistics. We profile the training time per local epoch and report the mean and standard deviation averaged over all clients and training rounds. Since GReg can be applied additively to most FL methods, we compare it primarily against approaches that modify either the federated procedure or the network architecture, namely HBN and FedWon. For HBN, we follow the official implementation from their codebase, while for FedWon, as no code was released, we re-implemented the method following the pseudo-code and details provided in their paper. All experiments are conducted on a single NVIDIA A6000 GPU. The results are summarized in Table 10. GReg introduces about a 1.45× training time overhead relative to FedAvg, which is lower than the 1.92× overhead of HBN and the 1.75× overhead of FedWon.

## G  USAGE OF LARGE LANGUAGE MODEL

Large Language Models (LLMs) were used as auxiliary tools in preparing this work. They assisted in polishing the manuscript by improving grammar, fluency, and readability. LLMs were also consulted for coding assistance, including drafting Python snippets, utility functions, and LaTeX formatting. All generated outputs were treated only as initial suggestions and were verified by the human authors. All research ideas and technical contributions are entirely those of the authors.

