# OpenReview forum: "Improving Batch Normalization in Federated Learning with Non-IID Features"
_ICLR.cc/2026/Conference — Submitted to ICLR 2026_

### Official Review · Reviewer_5r1v · 2025-10-19

**Soundness:** 3
**Presentation:** 3
**Contribution:** 3
**Rating:** 4
**Confidence:** 4

**Summary:**

This paper investigates why Batch Normalization (BN) underperforms in Federated Learning (FL) when clients have non-IID feature distributions (feature shift). The authors argue that global BN running statistics, averaged across heterogeneous clients, often mismatch any client’s true distribution, creating a train–test inconsistency at every BN layer. They propose *Local–Global Consistency Regularization (GReg)*: for each local batch, run two forward passes—one with local batch statistics and one with global running statistics—and add a symmetric KL penalty to align the two predicted distributions. A lightweight EMA smoothing of the global statistics on the server is used to stabilize training across rounds. Experiments on PACS, DomainNet, VLCS, and Camelyon17 report gains over FedAvg and several FL baselines, including improvements on unseen clients and across multiple BN-equipped CNN/ViT-like backbones.

**Strengths:**

**Originality:** Reframes BN failures in FL around feature non-IID (not only label skew) and introduces a simple, communication-free prediction-consistency regularizer between BN modes.

**Quality:** Uses explicit diagnostics (train/test loss gaps) to motivate the fix and shows consistent improvements across multiple datasets, FL algorithms, and partial-participation settings.

**Clarity:** Clear problem statement, method, and integration into FedAvg.

**Significance:** Practical for BN-based FL with pretrained CNNs; benefits on unseen clients suggest improved domain robustness.

**Weaknesses:**

**Limited architectural generality (BN-only):**
  The method is intrinsically tied to BN (it needs batch vs. running statistics). While the paper notes that BN remains “dominant in lightweight CNNs” and that pretrained ResNet backbones are widely used in FL, the claim of generality across architectures is overstated. Specifically:

  * No discussion or experiment on replacing all BN layers with LayerNorm or GroupNorm, which dominate in modern backbones such as ViTs, ConvNeXt, and Swin.
  * No attempt to generalize the consistency idea to non-BN norms (e.g., aligning affine parameters or normalized activations when running statistics do not exist).
    In practice, GReg applies only to BN-equipped models, limiting relevance for current state-of-the-art FL systems.

**Theory gap on the surrogate:**
  The assumed link between reducing the training-time gap and reducing the test-time gap is not formally established; the justification is largely empirical and dataset-specific.

**Statistical rigor:**
  Several reported gains are small and presented without standard deviations or confidence intervals; figures lack error bars and clear axes, so robustness is unclear.

**Computational oversight:**
  Two forward passes per batch effectively double local compute, yet runtime or FLOP overheads are not reported. The narrative emphasizes no extra communication cost but omits compute cost.

**Incomplete ablations and realism:**
  Missing ablations on EMA momentum, warm-up strategies for early rounds (when global stats are unstable), and small-batch regimes common in on-device FL. Feature-shift experiments assume balanced domains rather than more realistic heterogeneous client mixtures.

**Limited baselines and tuning parity:**
The comparison set omits several strong and relevant FL methods and BN-focused fixes (e.g.,  SCAFFOLD, FedDyn, MOON, FedNova), as well as domain-robust training baselines that could narrow the reported gains.

**Questions:**

* Can the local–global consistency idea extend to LayerNorm or GroupNorm (which lack running statistics), for example by aligning affine parameters or normalized activations?

* What is the actual runtime or FLOP overhead of the dual forward passes per batch? Please provide wall-clock comparisons.

* Can you report standard deviations or confidence intervals for the main tables and add error bars to figures to demonstrate statistical significance?

* How sensitive is performance to the regularization weight and the server-side EMA momentum? Would adaptive schedules help?

* How does GReg behave under mixed heterogeneity (both feature and label non-IID) and with more realistic, imbalanced client compositions?

* Can you include results on pure LayerNorm-based ViT backbones to verify the claimed architectural generality?

* To what extent is GReg acting as a general regularizer versus specifically correcting BN-statistic drift? Any disentangling experiments?

---

### Official Review · Reviewer_yT1L · 2025-10-20

**Soundness:** 4
**Presentation:** 2
**Contribution:** 2
**Rating:** 4
**Confidence:** 4

**Summary:**

- The field of non-IID Federated Learning is hyper-focused on the label-shift, which misleads researchers to believe batch norm shift is not important. This paper shows that when there is feature-shift (which is very likely in the real-world), batch norm shift becomes an important problem.
- The authors show the large difference in batch norm statistics comparing a label-shift and feature-shift setting (Figure 1) to motivate their method.
- To combat the batch norm shift, specifically the shift at train-test time in the batch norms, the authors use a KL-divergence regularization between the local and global batch norms to decrease the distance between the logits when using local batch norms and logits when using global batch norms.

**Strengths:**

- I appreciate the focus on feature-shift. The authors also included leave-one-domain-out, which is standard for domain generalization/adaptation testing, which is also appreciated.
- Good setup for the motivation of the problem using Figure 1 and Table 1. It shows the discrepancy of the batch norm and the harmful effects of the large discrepancy.

**Weaknesses:**

- The authors (correctly) claim that the non-IID field is too focused on label-shift, when we also have the feature-shift problem. However, the authors' results ONLY focus on feature shift. The paper would be made much more complete if they included results. The authors provide a comparison in Figure 1, yet they do not provide accuracies.
- Additionally, providing a figure similar to Figure 1 after training with the regularization would be very beneficial. Since the authors claim that their method should make the model more robust to the BN shifts, the actual statistics should reflect this.
- The paper would benefit from better framing of their approach. Reading for the first time, I was confused why this method would increase performance over FedProx, given that FedProx regularizes the weights (including batch norms) between local and global model. However, after much thought, it occurred to me that this paper is suggesting that their method makes the model INVARIANT to the batch norm. It does not, like FedProx does, try to decrease the distance between the global and local model weights.

**Questions:**

- Can the authors provide results for label-shift performance? It seems dishonest to claim that the field is too focused on label-shift, yet focus solely on feature-shift. I believe both are important.
- Can the authors provide the statistics for the batch norm shifts once their regularization is added?
- My third point in 'Weaknesses' also brings up another question. How do they authors believe the model would perform if batch norms were frozen or removed entirely? Additionally, would instance norms circumvent the batch norm shift?

---

### Official Review · Reviewer_MuVB · 2025-10-28

**Soundness:** 3
**Presentation:** 3
**Contribution:** 2
**Rating:** 2
**Confidence:** 4

**Summary:**

This paper addresses the challenge of non-IID feature distributions in federated learning (FL), specifically focusing on the limitations of batch normalization (BN) in such settings. The authors propose a **local-global consistency regularization** method that aligns global BN statistics during local training, allowing BN to be retained in the model. The proposed approach is evaluated on four datasets and demonstrates improvements over existing methods.

**Strengths:**

- The topic is relevant, as handling non-IID features remains a significant challenge in FL.
- The proposed method is conceptually straightforward and clearly presented.
- The paper considers both cross-device and cross-silo federated learning scenarios.

**Weaknesses:**

- The issue of inconsistency between training and testing BN statistics has been studied in prior works [1, 2, 3, 4]. The paper should further clarify its novel contributions in this regard—what specific insights or improvements does it offer beyond existing work?

- The use of the `loss gap’ to quantify inconsistency is simple and intuitive. However, relying solely on the loss value may not fully capture the nuances of the inconsistency. Are there alternative or complementary metrics that could offer a more comprehensive evaluation?

- The paper presents a bound on the loss gap. However, further analysis of how this bound relates to the training/generalization behavior would strengthen the theoretical contribution.

- While the paper critiques FedBN and SiloBN for not addressing cross-device FL, it does not experimentally evaluate these methods. Given their relevance, they should be included in the comaprison.

- In the generalization experiments, methods such as BN statistics adjustment[3, 4] and test-time adaptation[1] are not considered. Including these comparisons would provide a more thorough evaluation.

- The paper mentions the scenario of *stateful clients**, but does not include experiments involving unstable or intermittently participating clients. Evaluating the method under such realistic conditions are necessary.


References

[1] Wang, Dequan, et al. "Tent: Fully test-time adaptation by entropy minimization." *arXiv preprint arXiv:2006.10726* (2020).
[2] Li, Xiaoxiao, et al. "FedBN: Federated learning on non-IID features via local batch normalization." *arXiv preprint arXiv:2102.07623* (2021).
[3] Jiang, Meirui, et al. "UniFed: A unified framework for federated learning on non-IID image features." *arXiv preprint arXiv:2110.09974* (2021).
[4] Awais, Muhammad, Md Tauhid Bin Iqbal, and Sung-Ho Bae. "Revisiting internal covariate shift for batch normalization." *IEEE Transactions on Neural Networks and Learning Systems* 32.11 (2020): 5082–5092.

**Questions:**

-	How would be performance change when further decreasing \alpha? E.g., smaller than 0.05.

---

### Official Review · Reviewer_5jUR · 2025-11-02

**Soundness:** 2
**Presentation:** 3
**Contribution:** 2
**Rating:** 4
**Confidence:** 3

**Summary:**

This paper presents GReg, a simple regularization approach designed to address the inconsistency between local and global Batch Normalization (BN) statistics in feature-shift federated learning (FL). The paper first analyzes the performance gap between local and global BN statistics in FL models. Then, GReg is introduced to minimize the KL divergence between the outputs computed using these two statistics, thereby aligning their predictions. Experimental results demonstrate that GReg can be integrated with existing FL methods to yield performance improvements under feature-shift FL settings.

**Strengths:**

- The proposed GReg method is simple and easy to implement, making it readily applicable to existing FL approaches and capable of further improving their performance.

- The paper provides a theoretical bound on the loss gap between FL models using global and local batch statistics when applying GReg, making the effectiveness of the method theoretically well-grounded.

**Weaknesses:**

- At Lines 73-74 and 86-87, the paper argues that feature-shift FL is more challenging than label-shift FL because the variance of statistics across clients is larger in the feature-shift setting. However, larger variance alone does not necessarily imply that feature-shift FL is more difficult. The relative difficulty may depend on how these variances affect the loss landscape when switching between global and local BN statistics. For example, Table 1 shows that the performance gap between global and local statistics is smaller in feature-shift FL than in label-shift FL. Moreover, Table 2 indicates that FedAvg and other FL approaches already achieve performance close to centralized learning under feature-shift FL, whereas label-shift FL with highly skewed label distributions (e.g., determined by Dirichlet $\alpha$ or Shards) usually leads to a more substantial performance gap [1].



- The proposed GReg requires computing predictions from both FL models using global and local batch statistics in order to minimize their KL divergence, effectively doubling the forward-pass computation compared to FedAvg. This additional overhead may be undesirable in FL settings where client devices often have limited computational resources.

- The paper focuses on feature-shift FL. However, the motivation for specifically addressing feature-shift FL is not entirely clear. Based on Lines 88-89, the motivation seems to rely on the argument that feature-shift FL is more difficult than label-shift FL, which alone may not sufficiently justify the focus.

[1] 2023 NeurIPS Workshops Making Batch Normalization Great in Federated Deep Learning

**Questions:**

Besides the weakness shown in the above section, please also see the following questions:

Q1: At Lines 73-74 and 86-87, the paper argues that feature-shift FL is more challenging than label-shift FL. Following this perspective, can GReg also improve FL methods in label-shift settings? What is the performance of GReg under label-shift FL scenarios?

Q2: Table 1 shows that using local batch statistics can improve FedAvg. In addition, the positive loss gaps in Figures 2 and 3 indicate that using local batch statistics can result in lower loss compared to global statistics. This suggests that local batch statistics may be beneficial in feature-shift FL. How does this approach perform across different datasets, and how does it compare to GReg?

Q3: What is the additional computation cost introduced by GReg, and how does this cost trade off against the performance improvements shown in Table 2?

---

### Meta-Review · Area_Chair_abTK · 2025-12-23

**Summary:**

The paper initially received negative reviews, with scores of 4, 4, 4, and 2.

Reviewers identified several weaknesses in the paper, including a poorly supported argument that feature-shift FL is more challenging than label-shift FL, insufficient analysis of computational costs, and a lack of comprehensive baseline comparisons under realistic conditions. The authors did not provide a rebuttal to these concerns.

The area chair agrees with the reviewers' evaluation and recommends rejecting the paper.

**Reviewer Concerns:**

The authors did not respond to the reviewers' concerns.

**Reviewer Scores:**

The area chair expects the reviewers to maintain their initial scores.

---

### Decision · Program_Chairs · 2026-01-26

Reject